# A roadmap from the bond strength to the grain-boundary energies and macro strength of metals

Xin Li, Hao Wu, Wang Gao ✉ & Qing Jiang

Correlating the bond strength with the macro strength of metals is crucial for understanding mechanical properties and designing multi-principal-element alloys (MPEAs). Motivated by the role of grain boundaries in the strength of metals, we introduce a predictive model to determine the grain-boundary energies and strength of metals from the cohesive energy and atomic radius. This scheme originates from the d-band characteristics and broken-bond spirit of tight-binding models, and demonstrates that the repulsive/attractive effects play different roles in the variation of bond strength for different metals. Importantly, our framework not only applies to both pure metals and MPEAs, but also unravels the distinction of the bond strength caused by elemental compositions, lattice structures, high-entropy, and amorphous effects. These findings build a physical picture across bond strength, grain-boundary energies and strength of metals by using easily accessible material properties and provide a robust method for the design of high-strength alloys.

It has been a long-term goal to directly correlate bond strength with the strength of metals, aiming to access the upper bound of strength. With the growing demand for high-performance materials, increasingly complex alloys are being studied and applied, such as the multi-principal-element alloys (MPEAs) in polycrystalline and amorphous states[1–6]. However, the structure-property relationships between bond strength and macro-mechanical properties across different metallic systems pose significant challenges because of the vast composition spaces and complex structures of metals, particularly MPEAs. These obstacles necessitate the development of simple and effective physics-based descriptors to facilitate the material design.

The origin of the strength of metals was explored by relating to the bond-strength deriving materials properties including heat of fusion or stacking fault energies, by considering grain-boundary (GB) sliding as viscous flow or the relation between stacking fault energies and the splitting distance of partial dislocations[7–9]. However, the application of these models is mainly limited to pure metals. Although some descriptors like the density-functional-theory (DFT) determined interstitial electron density $\rho_0$[10–12] were further proposed as a measure for the strength of alloys[13], there is still a lack of bond-strength

descriptors to directly correlate the macro properties. On the other hand, the ultimate strength of metals generally depends on the grain size and thus relates to the properties of GBs[9,13–16]. To improve the mechanical properties of metals, determining the properties of GBs by using intrinsic properties of metals has been widely studied[17–24], as referred to GB engineering[25]. Although some machine-learning approaches are effective in determining the GB properties like GB energies[26,27], they generally provide less insight into physical meanings and do not contain concise mathematical functions. These black-box methods also require effective features and a large amount of data. Moreover, these studies for the determinants of GB properties are generally separated from those for the macro strength. There is still a lack of the universal quantification of the atomic-scale bond strength, which can be used to understand the properties caused by atomic movement and lattice friction, and thus to determine the macro mechanical properties for both pure metals and alloys.

In this contribution, we propose intrinsic descriptors, which directly originate from bond strength, to determine the strength of metals by combining the cohesive energy and atomic radius. Our descriptors build a predictive framework of the GB energies for face-

Key Laboratory of Automobile Materials, Ministry of Education, Department of Materials Science and Engineering, Jilin University, 130022 Changchun, China.
✉ e-mail: wgao@jlu.edu.cn

centered-cubic (FCC), hexagonal-close-packed (HCP), and body-centered-cubic (BCC) metals and a comprehensive physical picture across the bond strength, GB energies, and strength of metals. Moreover, these descriptors with a rule-of-mixture (RoM) estimate can be applied to the MPEAs in polycrystalline and amorphous states. Our descriptors are motivated by d-band properties and broken-bond mechanism and correlated with the classical theories of modulus, reflecting their solid physical basis. These findings are crucial to understanding and predicting the strength properties of metals.

## Results

The well-known mechanism of the Hall-Petch relationship for polycrystalline metals[28,29] shows that the dislocations glide at the grain boundary, leading to the GB sliding and building up the stresses at the triple junctions of GBs[15,16]. These stresses emit the dislocations to pile up other GBs, causing the other GB movement and thus the deformation of metals. The maximum shear strength emerges as the grain size decreases below a critical grain size, accompanied by the strengthening mechanism transforming from intragranular deformation to intergranular deformation[9,30]. For the pure metals, the driving forces for the GB sliding and migration mainly derive from two parts: the stored deformation energies and grain boundary energies[31]. The formers are caused by the density and energies of defects (e.g. dislocations); however, the grains below the critical size are too small to accommodate dislocation movement[9]. Therefore, the motion of GBs comes under the spotlight. The stability of GBs (namely the GB energies) thus dominates the strain rate and the deformation degree. We thus try to determine macro properties (e.g. the maximum shear strength) of metals by quantifying the energetics of GBs from the bond properties of materials.

### GB energies of pure metals

Generally, one uses the cohesive energy $E_{coh}$ to characterize the bond properties of a metal, which is expressed according to the tight-binding models as[32]:

$$E_{coh} = \left(\frac{q}{p} - 1\right) W_d \frac{N_d(N_d - 10)}{20} \quad (1)$$

Here, $N_d$ and $W_d$ are the d-band occupation and d-band width, which are defined as the zeroth moment and the square root of the second moment of the d-band density of states, respectively. $q/p$ represents the ratio of attractive and repulsive parts. However, the atoms at defects, such as the surface or GB, have a different coordination compared with the bulk atoms, which produces a variation of the defect local density of states (LDOS), leading to the modification of the d-band width[33,34].

Inspired by the potential correlation between surface energy and GB energy ($\gamma_{surf} \propto E_{GB}$)[35], we start from the surface energies by following the tight-binding models to understand the determining factors of GB energies. In the framework of the tight-binding models, the surface energy is deduced according to the bond-cutting models as[32],

$$\gamma_{surf} = \left[1 - \frac{q}{p}\left(\sqrt{\frac{CN_S}{CN_B}} + 1\right)\right] \left(\sqrt{\frac{CN_S}{CN_B}} - 1\right) W_d \frac{N_d(N_d - 10)}{20} \quad (2)$$

Here, $CN_S$ and $CN_B$ are the coordination number of surface and bulk respectively. The coordination number of a given atom at surfaces (or GBs) is the sum of the weights of its nearest neighbors (that is obtained by partitioning their bond length with the bond length of bulk). The nearest neighboring atoms are classically identified by the collision detector of spheres, which calculates the pairs of spheres that overlap for a given atom. The test allows us to adopt the cutoff radius being 1.1 times of the shortest atomic distance in a pristine bulk structure, considering the deformation of the surface (or GB) region

compared with the bulk. These approaches are widely used in determining the coordination number.

If neglecting the repulsive contribution of surface energy ($q/p = 0$), the attractive contribution derives the square-root broken-bond model as $\gamma_{surf} \propto CN^{1/2}E_{coh}$. If the attractive (band) contribution of surface energy $E_{band}$ follows the behavior of the repulsive contribution of surface energy $E_{rep}$ as $CN_S$ changes (corresponding to the condition that the surface is unrelaxed), one obtains the broken-bond model as $\gamma_{surf} \propto CNE_{coh}$. Usually, $\gamma_{surf} \propto CNE_{coh}$ performs better in strong chemical bond materials, whereas $\gamma_{surf} \propto CN^{1/2}E_{coh}$ does better in weak chemical bond materials, as the strong bonds generally undergo smaller relaxation than the weak bonds towards the bond breaking.

GBs are composed of two surfaces, the resulting chemical bonds at the interfaces certainly experience tension or compression. Different from the bond properties of surfaces, the interface bonds are most likely affected by the size of matrix atoms. Therefore, one should consider both the bond energies and the size effects at GBs, which influence the GB energies and thus the macro properties related to the GBs. According to the tight-binding models, one can readily derive $W_d$ with respect to the change of the bond length at the GB interfaces as[32,36]

$$W_d = 115\hbar^2 R_d{}^3/(mL^5) = 6.83\hbar^2 R_d{}^3/(mr_0{}^5) \quad (3)$$

where $L$ is the nearest neighbor distance, $m$ is the mass of an electron, $R_d$ is the spatial extent of the d-orbitals, and $r_0$ is the atomic radius. Namely $W_d \propto 1/r_0{}^5$. The dangling bonds of surfaces are partially saturated at the GB interfaces, generating the coordination number of the atoms at the GB interfaces ($CN_{GB}$). One can thus reasonably deduce $W_d \propto 1/r_0{}^5 \propto CN_{GB}$ for the atoms at the GB interfaces. We now propose two expressions to determine the GB energies. One is,

$$E_{GB} \propto \gamma_{surf} \propto E_{coh}CN_{GB} \propto E_{coh}/r_0{}^5 \quad (4)$$

for the strong bond metals with a minor bond relaxation according to the broken-bond model. The other is,

$$E_{GB} \propto \gamma_{surf} \propto E_{coh}CN_{GB}{}^{1/2} \propto E_{coh}/r_0{}^{2.5} \quad (5)$$

for the relatively weak bond metals with a significant bond relaxation according to the square-root broken-bond model. For the convenience of practical application, we adopt the experimental values of the cohesive energy $E_{coh}$ and atomic radius $r_0$[37,38], which are easily obtained by table looking up. The cohesive energy is defined as the energy required to arrange the individual atoms into a crystalline state at 0 K and 1 atm. The radius of an atom X is defined as half the length of a X-X bond in the most stable crystal or molecule structure. The values of $E_{coh}$ and $r_0$ of different elements are shown in Supplementary Table S1.

We now study the performance of our models in determining the GB energies[19], by considering 30 transition metals (TMs) and 11 main-group metals (MGMs, containing Li, Na, K, Rb, Cs, Be, Mg, Ca, Sr, Ba and Al). Starting from BCC metals, BCC structures exhibit distinct bond strength, with the cohesive energy >4.2 eV/atom for BCC TMs (4.2 - 8.9 eV/atom) and <1.9 eV/atom for BCC MGMs (0.80 - 1.90 eV/atom). We find that the GB energies can be better described by $E_{coh}/r_0{}^5$ for BCC TMs but by $E_{coh}/r_0{}^{2.5}$ for the BCC MGMs (Fig. 1a, b and Supplementary Fig. 1), regardless of tilt or twist GBs and the GB orientations[19], which are in good agreement with the prediction of our model (Eqs. 4 and 5).

We turn to study the 13 HCP metals, most of which are TMs (10 TMs vs 3 MGMs) with cohesive energy close to the BCC TMs (3.90 - 8.17 eV/atom). Accordingly, the GB energies[19] scale better with $E_{coh}/r_0{}^5$ instead of $E_{coh}/r_0{}^{2.5}$ for 13 HCP metals in Fig. 1c, d and Supplementary Fig. 2a and b as our scheme predicted, which reproduce

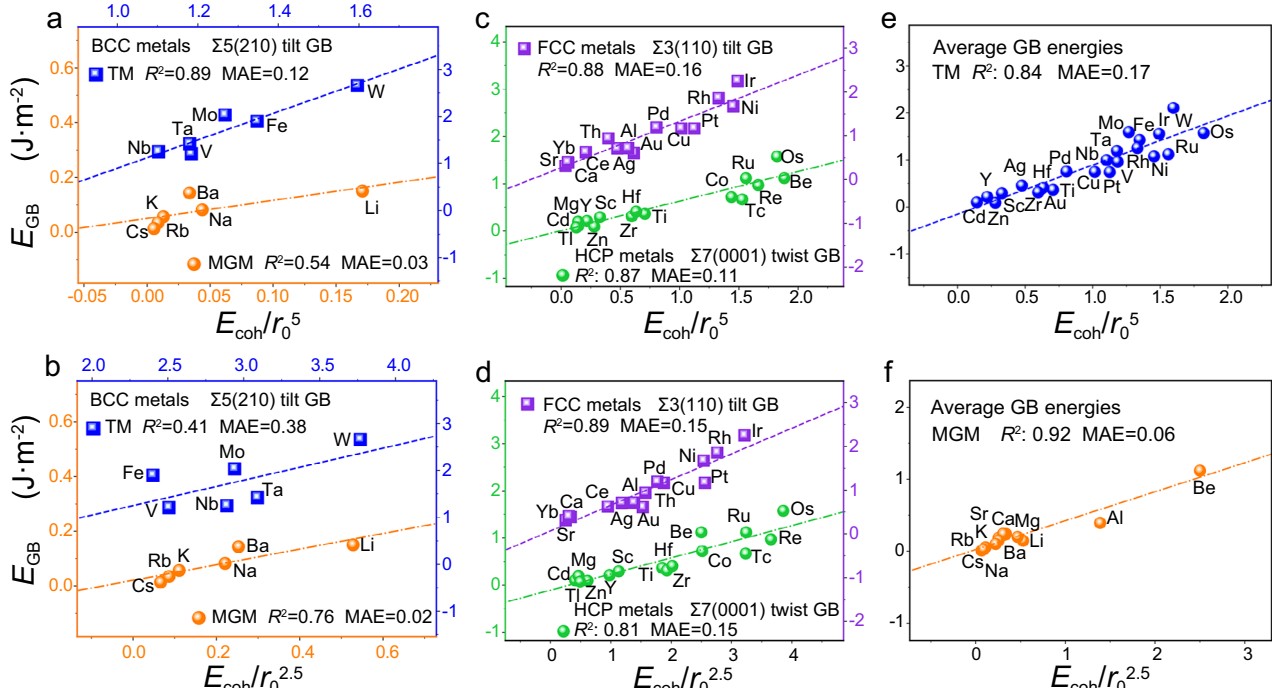

**Fig. 1 | Grain-boundary (GB) energies ($E_{GB}$) as the functions of our descriptors[19].** GB energies with (**a**) $E_{coh}/r_0^5$ and (**b**) $E_{coh}/r_0^{2.5}$ for body-centered-cubic (BCC) metals. The blue squares and orange dots represent BCC transition metals (TMs) and main-group metals (MGMs). GB energies with (**c**) $E_{coh}/r_0^5$ and (**d**) $E_{coh}/r_0^{2.5}$ for hexagonal-close-packed (HCP, green dots) and face-centered-cubic (FCC, purple squares)

metals. The average GB energies with (**e**) $E_{coh}/r_0^5$ for TMs (blue dots) and (**f**) $E_{coh}/r_0^{2.5}$ for MGMs (orange dots). The accuracy is measured by mean absolute error (MAE) and regression coefficient ($R^2$). All the dashed lines are obtained from linear fitting. Source data are provided as a Source Data file.

the behavior of BCC counterparts. In the case of FCC metals (15 metals), the cohesive energy gradually increases from 1.6 eV/atom to 6.94 eV/atom and overall is relatively intermediate compared to that of BCC and HCP metals. $E_{coh}/r_0^{2.5}$ and $E_{coh}/r_0^5$ are comparable in describing the GB energies for the studied 5 tilt GBs and 4 twist GBs of the 15 FCC metals[19] (Fig. 1c, d, and Supplementary Fig. 2c–f), which is also compatible with the prediction of our scheme.

Besides the symmetric tilt and twist GBs, our framework can be also extended to describe the GB energies of complex asymmetric GBs, such as (221)/(110), (411)/(100), (710)/(110) and (1130)/(970) GBs of FCC and BCC metals (Supplementary Fig. 3. The structures of the GBs are shown in Supplementary Fig. 4). Moreover, we also study more intricate triple-junction GBs, which can be commonly found in polycrystals. Our descriptor also shows a linear relationship with the strengthening energy (which is another property related to GB stability) of Σ3(111)-Σ3(111)-Σ9(221) triple-junction GBs (Supplementary Fig. 3f). All these results indicate that our descriptors are applicable to determine the GB stability regardless of the GB types.

Most metals are actually used in a polycrystalline form instead of a single-crystal form. The calculated amorphization energies also correspond to the average energy of high-angle GBs[9,31]. In other words, the average of the GB energies likely better presents the stability of GBs of real metals. We thus correlate our descriptors with the average GB energies of different tilt and twist GBs for BCC, FCC, and HCP metals, finding that $E_{coh}/r_0^5$ can determine the average GB energies of TMs whereas $E_{coh}/r_0^{2.5}$ can describe the average GB energies of MGMs well as illustrated in Fig. 1e, f. Recalling that the previous descriptors $E_{coh}/a_0^2$, $a_0C_{44}$, and $a_0G$ ($C_{44}$ and $G$ are the elastic constant and shear modulus respectively) were proposed to determine the GB energy scaling slopes between metals for different lattice structures[21,39,40], our descriptors show the better accuracy and consistency in directly determining the GB energies for both TMs and MGMs regardless of the lattice structures. Importantly, our results uncover the different

behavior of GB stability for TMs and MGMs: $E_{coh}/r_0^5$ applies to the metals with strong bonds and large cohesive energies (mainly TMs) while $E_{coh}/r_0^{2.5}$ is applicable to those with weak bonds and small cohesive energies (mainly MGMs).

We further discuss the physical origin of our descriptors from the perspective of electronic effects, by correlating our descriptors with the average interstitial electron density $\rho_0$ generated from the properties of a homogeneous electron gas in DFT[10–13]. $\rho_0$ is defined as the charge $Q_0$ per unit volume $\Omega_0$ at the interstitial around an atom as $Q_0/\Omega_0$ and has been suggested as a measure of the bond strength of metals[13]. Notably, the descriptor $E_{coh}/r_0^5$ scales linearly with $\rho_0$ for the TMs, whereas $E_{coh}/r_0^{2.5}$ describes the $\rho_0$ of the MGMs more accurately (Fig. 2a). The correlations of $\rho_0$ vs $E_{coh}/r_0^5$ for TMs and $\rho_0$ vs $E_{coh}/r_0^{2.5}$ for MGMs further demonstrate that the bond properties of metals with strong bonds are distinguished from those with weak bonds. On the one hand, the more interstitial electrons per unit volume (the larger $\rho_0$ and the stronger bond), the more difficult atoms get close to each other, increasing the difficulty of compressing the electron gas and causing the increase of the repulsion term of the bond energy[13]. This behavior corresponds to the fact that the stronger bond is harder compressed. As the TMs generally have small radii and strong bonds, the repulsive contribution is significant in the bond relaxation, echoing the relatively small bond relaxation and the dominant role of $E_{coh}/r_0^5$ in the GB energy of TMs. On the other hand, the fewer interstitial electrons per unit volume (the smaller $\rho_0$ and the weaker bond), the more space for interatomic compression. As the MGMs generally have large radii and weak bonds, the attractive contribution is significant in the bond relaxation, echoing the significant bond relaxation and the dominant role of $E_{coh}/r_0^{2.5}$ in the GB energy of MGMs. Overall, our descriptors together with the $\rho_0$ characteristics and broken-bond mechanism consistently demonstrate that the rule of the bond strength of metals, which is essential in determining the behavior of GB stability, changes from TMs to MGMs significantly.

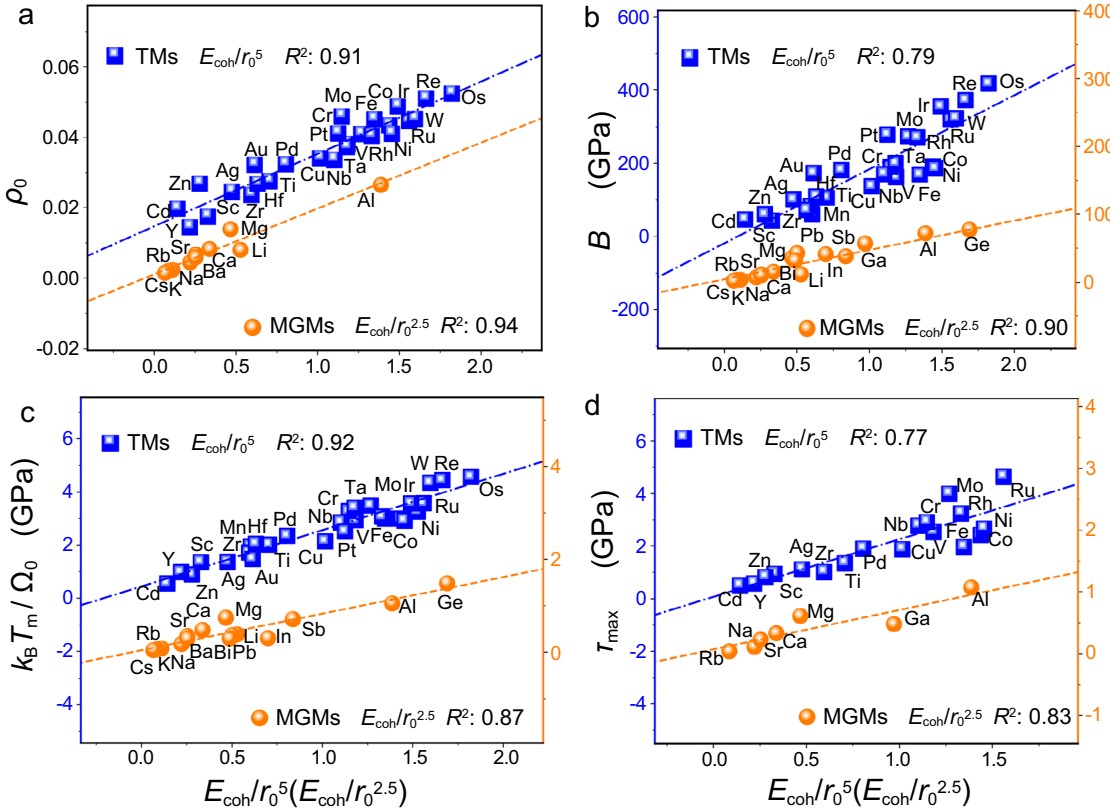

**Fig. 2 | The correlation between our descriptors, the previously proposed characteristics and the macro properties of different transition metals (TMs, blue squares) and main-group metals (MGMs, orange dots)[9,13,37]. a** The interstitial electron density $\rho_0$. **b** The bulk moduli $B$. **c** The average heat of fusion $k_B T_m/$ $\Omega_0$. **d** The maximum shear strength $\tau_{max}$. The accuracy is measured by regression coefficient ($R^2$). All the dashed lines are obtained from linear fitting. Source data are provided as a Source Data file.

## Macro-mechanical properties for pure metals

We now use our descriptors to determine the modulus of metals which are generally accepted to depend on the bond strength. Figure 2b shows that $E_{coh}/r_0^5$ is accurate for the bulk modulus $B$ of TMs and $E_{coh}/$ $r_0^{2.5}$ is more accurate for the MGMs with the regression coefficients of >0.8. Similar linear correlations can be found between Young's modulus and shear modulus of metals and our descriptors. Our findings overcome the limitations of the previously proposed volumetric cohesive energy descriptor $E_{coh}/\Omega_0$ in estimating the modulus of diverse metals[41], further demonstrating the distinction of the metals with large and small bond strength. Modulus refers to the ratio of stress to strain with the applied stress or the deformation degree of a material. The stronger bond energy leads to a weaker ability of bond relaxation, indicating a stronger resistance to deformation and a larger bulk/Young's modulus.

Our descriptors not only provide a physical insight into the bond strength of metals by combining the bond energy and bond length, but also exhibit good compatibility with the classical theories of modulus. In these theories, the bulk modulus is related to the force-volume ($\sigma$-$\Omega$) or the potential energy-volume ($U$-$\Omega$) relationship as[7],

$$B = -\Omega_0 \left(\frac{\partial \sigma}{\partial \Omega}\right)_{\Omega_0} = \Omega_0 \left(\frac{\partial^2 U}{\partial \Omega^2}\right)_{\Omega_0} \quad (6)$$

Here, $\Omega_0$ is the equilibrium volume, corresponding to the equilibrium interatomic spacing, simply the atomic radius $r_0$. The potential energy between two atoms ($u_{ij}$) can be expressed as[7]

$$u_{ij} = -c_1 r^{-n} + c_2 r^{-m} \quad (7)$$

Here $n$ and $m$ correspond to the attractive and repulsive parts. For the ionic solids and some alkali metals, the attractive energy is well represented by a simple coulombic interaction with $n = 1$ and $m$ is empirically >6. The cohesive energy is determined primarily by the attractive energy ($E_{coh,\,attr} \propto r_0^{-1}$) at the equilibrium conditions. According to the relationship $B \propto E_{coh}/r_0^{2.5}$ for MGMs, one can obtain $B \propto E_{coh,attr}/r_0^{2.5} \propto r_0^{-3.5}$. In practice, Supplementary Fig. 5a shows that $E_{coh}$ of MGMs exhibits a function of $r_0^{-1.6}$ for MGMs, leading to the $B \propto E_{coh,\,attr}/r_0^{2.5} \propto r_0^{-4.1}$, in consistent with the classical models as $B \propto r_0^{-4}$. From the perspective of atomic interaction force $\sigma$, the repulsive and attractive forces are equal at equilibrium. However, the classical model only considers the contribution of the attractive force, and thus derives the relationship $B \propto -\Omega_0 \left(\frac{\partial \sigma}{\partial \Omega}\right)_{\Omega_0} \propto (\Omega_0)^{-4/3} \propto r_0^{-4}$. These results demonstrate that the attractive force dominates the bond strength in determining the modulus of MGMs.

In the case of TMs, the contribution of attractive and repulsive parts of the potential energies generally can be regarded as the function of $r_0^{-4}$ and $r_0^{-6}$ ($n = 4$ and $m = 6$, $E_{coh,\,attr} \propto r_0^{-4}$ and $E_{coh,\,rep} \propto r_0^{-6}$) according to the tight-binding theories[32]. From the perspective of bond energy, on the basis of the linear relationship $B \propto E_{coh}/r_0^5$ for TMs, it naturally generates $B \propto E_{coh,attr}/r_0^5 \propto r_0^{-9}$ as the attractive effects dominate the cohesive energies at the equilibrium conditions. Indeed, the bulk moduli linearly correlate with $r^{8.8}$ for TMs ($-r_0^{-9}$) in experiments, as shown in Supplementary Fig. 5b. Although the repulsive and attractive

forces are equal at the equilibrium conditions, the former plays a more important role in TMs than in MGMs as the bond length of TMs is obviously smaller than that of MGMs. Therefore, one can also derive $B \propto -\Omega_0 \left(\frac{\partial \sigma}{\partial \Omega}\right)_{\Omega_0} \propto r_0^{-9}$ for TMs if only considering the contribution of the repulsive force.

Overall, the resulting decay behavior from both the energy and force derivations are consistent, regardless of MGMs or TMs, demonstrating the conclusion that the attractive and repulsive force dominates the trend of the modulus of MGMs and TMs, respectively. All these results show that our descriptors, derived from tight-binding models, are not only compatible with the classical theories, but also uncover a physical picture that the variation of bulk moduli of different metals originates from the repulsive effects of metals with strong bond ability like TMs but from the attractive effects of metals with relatively weak bond ability like MGMs.

It's well known that the bond strength of a metal can be intuitively reflected by the melting temperature $T_m$. The heat of fusion $L$, $L = k_B T_m$, was used to assess the bonding energy, which also provides a rough estimate of the high-angle GB energies[7,9,13]. The Young's moduli, correlated with stiffness, can be expressed as $100 k_B T_m / \Omega_0$. We find that $k_B T_m / \Omega_0$ can be better determined by the descriptor $E_{coh}/r_0^5$ for TMs but by $E_{coh}/r_0^{2.5}$ for MGMs (in Fig. 2c), further demonstrating the applicability of our descriptors in determining the bond strength and moduli. Our descriptors can describe other GB-dependent strength measures well, such as the maximum shear strength $\tau_{max}$ as illustrated in Fig. 2d[9,14]. Similar to the GB energy and modulus, $E_{coh}/r_0^5$ scales linearly with $\tau_{max}$ for all the studied metals, especially for TMs, and $E_{coh}/r_0^{2.5}$ describes the $\tau_{max}$ of the MGMs with small cohesive energies more accurately. These results demonstrate the predictive capability of our descriptors in determining the GB-dependent strength of pure metals.

### The framework for alloys

We further show the application of our descriptors in alloys, especially for the high-entropy alloys (HEAs, a type of MPEAs) with commonly

more than four elements and random atomic distribution, and for the bulk-metallic glasses (BMGs) with amorphous structures (in the form of the binary alloys and MPEAs). Therefore, one needs to consider the effects of elemental compositions with different concentrations and the structural difference among pure metals, HEAs and BMGs in order to describe the macro-properties of alloys. Here, we extend our descriptors to alloys, by combining with the rule-of-mixture (RoM) estimate, that is, the average of the individual composition together with the weight of their concentration as

$$(E_{coh}/r_0^5)_{RoM} = \sum_{i=1}^{N} c_i E_{coh,i}/r_{0,i}^5 \qquad (8)$$

$$(E_{coh}/r_0^{2.5})_{RoM} = \sum_{i=1}^{N} c_i E_{coh,i}/r_{0,i}^{2.5} \qquad (9)$$

Here, $N$ is the number of elements in the alloys, $c_i$ is the concentration of the $i$th elements and $E_{coh,i}$ and $r_{0,i}$ are the cohesive energy and atomic radius of the $i$th elements. Notably, the actual atomic volume of a given element varies in different alloys. For example, HEAs may have mixture phases (such as BCC + FCC, FCC+intermetallics), and BMGs exhibit complicated geometric structures due to structural disorder, making the radius choice based on the alloy phases not unique. Nevertheless, the difference of adopted atomic sizes in different phases cause negligible effects on the predictive power of our descriptors (Fig. 3a and Supplementary Fig. 7a). Therefore, we simply use the cohesive energies and atomic radii obtained from the most stable crystal or molecule structure, which are easily obtained by table looking up.

### Macro-mechanical properties of HEAs

In the case of HEAs, Fig. 3a shows that $(E_{coh}/r_0^5)_{RoM}$ exhibits a linear correlation with Young's modulus for Al-Co-Cr-Cu-Fe-Mn-Ni based 3 d

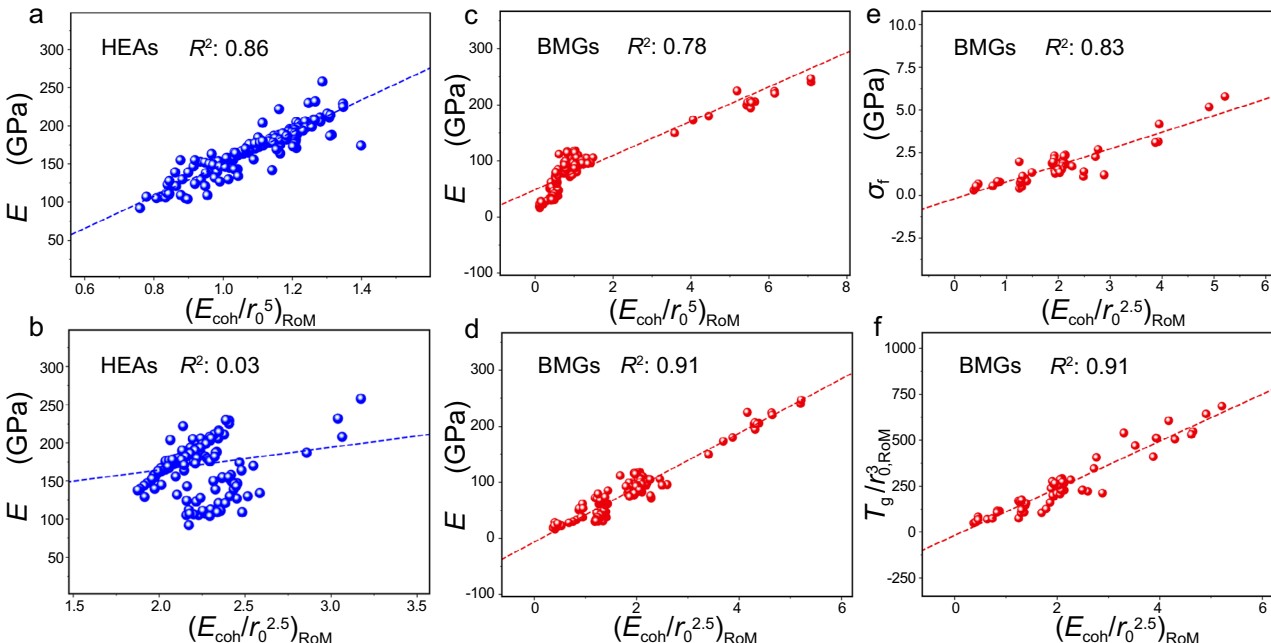

**Fig. 3 | Macro properties of alloys as the functions of our descriptors[5,42].** Young's moduli $E$ with (**a**) $(E_{coh}/r_0^5)_{RoM}$ and (**b**) $(E_{coh}/r_0^{2.5})_{RoM}$ for Al-Co-Cr-Cu-Fe-Mn-Ni-based 3 d high-entropy alloys (HEAs) and refractory-metal-based HEAs. $E$ with (**c**) $(E_{coh}/r_0^5)_{RoM}$ and (**d**) $(E_{coh}/r_0^{2.5})_{RoM}$, (**e**) Fracture strength $\sigma_f$ with $(E_{coh}/r_0^{2.5})_{RoM}$, and (**f**) the descriptor $T_g/r_{0,RoM}^3$ with $(E_{coh}/r_0^{2.5})_{RoM}$ for Zr-, Cu-, Ti-, Mg-, Fe-, and alkaline-earth-based bulk-metallic glasses (BMGs, in the form of binary and multiple-

elemental alloys). The accuracy is measured by regression coefficient ($R^2$). The blue and red circles represent HEAs and BMGs, respectively. All the dashed lines are obtained from linear fitting. The corresponding alloy composition of each data-point can be found in the Source data file. Source data are provided as a Source Data file.

HEAs and refractory-metal-based HEAs[42], with the regression coefficient of 0.86. The descriptor $(E_{coh}/r_0^5)_{RoM}$ is more accurate than the DFT-calculated $\rho_0$ in determining Young's modulus (the regression coefficient of ~0.7 for only BCC HEAs[13]). Notably, there is no universal relation between Young's modulus and $(E_{coh}/r_0^{2.5})_{RoM}$ with RoM across different types of HEAs (Fig. 3b). The components of these HEAs are mainly TMs, with only a small amount of one or two main-group-element components like Al, Si, and Sn. These results further recall that $E_{coh}/r_0^5$ is mainly applicable to TMs with the strong bonding ability. Moreover, the correlation of $(E_{coh}/r_0^5)_{RoM}$ and measured Young's modulus for HEAs can be applied across various lattice structures, containing FCC, BCC, FCC + BCC, and some intermetallics. Our findings indicate that Young's modulus of the HEAs composed of metals with the strong bond ability are mostly affected by the repulsive effects of bonding. Moreover, the macro properties like Young's modulus are mainly influenced by the bond strength of the elemental compositions and the corresponding concentrations, instead of the types of lattice structures. These results indicate that the bond strength of HEAs experiences the mean-field effects in determining the modulus.

In addition, the linear function of modulus and $(E_{coh}/r_0^5)_{RoM}$ can provide physical guidance for designing new alloys. Specifically, increasing the concentration of the elements with large descriptors $E_{coh}/r_0^5$ in HEAs may improve the modulus of the material, while the addition of the elements with small descriptors may decrease the modulus. For example, the modulus of $Al_{0.5}CoCrCuFeNi$ and CoCr-CuFeNi alloys decreases with the increase of Ti solute, since Ti exhibits small descriptor compared with the matrix elements Co, Cr, Cu, Fe and Ni, which have been proven experimentally[43–45]. The additions of Tc, Re, Os, and Ni with large descriptor values also enhance the Young's modulus of CoAlV, CoVTa, and CoVTi alloys[46–48]. All these experimental and theoretical results verify the validity and application of our framework in determining the macro-mechanical properties of HEAs.

### Macro-mechanical properties for BMGs

Interestingly, we find that our descriptors are also suitable in determining the strength properties of BMGs, although BMGs have no lattice structures and GBs compared to HEAs. Figure 3d illustrates that the measured Young's modulus $E$ of BMGs[5] is a good linear function of $(E_{coh}/r_0^{2.5})_{RoM}$, across various binary and multiple-principle-element BMGs such as Zr-, Cu-, Ti-, Mg-, Fe-, and alkaline-earth-based BMGs, with the regression coefficient of >0.9. In contrast, for the same measured Young's modulus, $(E_{coh}/r_0^5)_{RoM}$ exhibits a significant deviation (~100% at the descriptors with small values) in prediction (Fig. 3c). In addition, $(E_{coh}/r_0^{2.5})_{RoM}$ is also effective in determining the measured fracture strength $\sigma_f$ and shear modulus $G$ of BMGs[5] (Fig. 3e and Supplementary Fig. 6a). These results suggest that the considered BMGs follow similar bonding rules regardless of the change in elemental compositions and concentrations. Clearly, the validity of $(E_{coh}/r_0^{2.5})_{RoM}$ (instead of $(E_{coh}/r_0^5)_{RoM}$) implies the dominant role of the attractive force of BMGs in determining the modulus of BMGs according to our scheme, which essentially originates from the structural properties of BMGs. BMGs have lower mass density compared to their crystalline counterpart, as the atomic bond lengths and angles vary greatly and some atoms are even unbonded with their neighbors in BMGs[49]. The significant bond relaxation of BMGs enlarges the interstitial atomic space and weakens the bond strength, making TM and MGM elements behave consistently in the amorphous states with a law of $(E_{coh}/r_0^{2.5})_{RoM}$.

Our descriptor can also estimate the glass-transition temperature $T_g$ of BMGs, which reflects the bond strength for BMGs as the melting temperature $T_m$ for crystalline alloys. The corresponding heat of fusion can be expressed as $k_B T_g/\Omega_0$, similar with the $k_B T_m/\Omega_0$ for crystalline alloys. We find that $T_g/r_{0,RoM}^3$ scales with $(E_{coh}/r_0^{2.5})_{RoM}$ as illustrated in Fig. 3f. Notably, $T_g$ data are experimentally measured and only $r_0$ in $T_g/r_{0,RoM}^3$ experience the RoM estimate, while $E_{coh}/r_0^{2.5}$, as a single descriptor for each element, is averaged with the weight of concentration in $(E_{coh}/r_0^{2.5})_{RoM}$ to apply to BMGs. In contrast, the previously proposed descriptor $T_g/\Omega_{0,RoM}$ (here only $\Omega_0$ with RoM estimate) are limited in determining the moduli of BMGs, since it has no universal linear relationships with the moduli across different types of BMGs, especially for the BMGs with the rare-earth metals (Supplementary Fig. 6b).

Notably, $T_g$ is usually taken as the material characteristic of BMGs, with a weak dependence on cooling rates[5]. In addition, the cooling rates through the preparation process affect the macro properties of metallic glasses to some extent. However, the adopted BMGs are usually obtained under relatively low cooling rates (on the order of $<10^3$ k/s), reaching the millimeter level of >3 mm, which have a minor influence on $T_g$, Young's modulus and fracture strength[5,50,51]. The studied properties of BMGs under these conditions are thus regarded as the material characteristics. The kinetic properties such as cooling rate effects need to be further studied in the future.

For the pure metals, with the decrease of the grain size, the number of atoms at GBs increases and thus the metal structures tend to be disorder. Accordingly, the bond density of the metal reduces, leading to the decrease of the interatomic bond strength ($E_{coh}$). Therefore, the value of the descriptor ($E_{coh}/r_0^5$) turns to decrease and thus the modulus gets smaller according to our framework, which is consistent with the previous findings[52].

### Discussion

Overall, our framework is applicable to both simple and complicated systems, such as from simple symmetric GBs to complex asymmetric GBs and triple-junctions, and from pure metals to HEAs and BMGs. These results suggest that the melting/glass-transition temperature, modulus and shear strength all derive from a sole physical origin, the bond strength, which can be well determined by our descriptors. Our descriptors are derived from the d-band characteristics and broken-bond mechanism of tight-binding models, and are compatible with the classical theories of bulk modulus. The power law of the $r_0$ in our descriptors is physics-based and certain: 5 for TMs and HEAs, and 2.5 for MGMs and BMGs. The former corresponds to the metals with strong bonds, while the latter applies to those with weak bonds.

Our descriptors are more accurate than the previous ones. For the prediction of GB energies, the regression coefficient is 0.84 for our descriptors compared with 0.76 for $\rho_0$ (Supplementary Fig. 7b). For the prediction of the macro strength of HEAs, the regression coefficients are 0.86 for our descriptors (Fig. 3a and c), but 0.67 for $\rho_0$ (see Figure 5 of ref. 13) and 0.74 for $\rho_0/\Omega$ (Supplementary Fig. 7c). Moreover, $\rho_0$ requires time-burdensome numerical calculations, the data of which are still unavailable for some elements such as C, Si and rare-earth elements. In comparison, the values of cohesive energy and atomic radius in our descriptors cover almost all the elements, and are all accessible by table looking up, which reduces the numerical uncertainty. From the view of physical pictures, compared with $\rho_0$ for understanding the moduli of metallics from the electronic perspective, our descriptors directly start from bond strength to correlate the macro properties from the view of atomic-level bonding. The proposed descriptors also unravel a physical picture between bond strength and macro strength: the repulsive effects play the dominant roles in the variation of the strength of metals with strong bond strength, such as TMs and HEAs, while the attractive effects dominate the variation of the strength of metals with the weak bond strength and the large atomic relaxation, like MGMs and BMGs.

Moreover, our descriptors provide physics-based features for machine learning. Supplementary Fig. 8 shows that the machine-learning models, based on the support-vector machine, random-forest regression, and gradient boosting decision tree algorithm, can accurately determine the modulus of HEAs and BMGs, by combining with only one descriptor $E_{coh}/r_0^5$ (for HEAs) or $E_{coh}/r_0^{2.5}$ (for BMGs). But

these machine-learning models cannot be extracted any concise mathematical function. Notably, it's known from the learning curves (Supplementary Fig. 8g, h) that with 20%-data training, the scores on the test sets are >0.85. Obviously, a small part of the data can determine the modulus efficiently. All these results further demonstrate the simplicity and advantage of our physical-based descriptors in determining the strength-related properties. The solid physical origin of our framework also contributes to addressing the drawback of black-box machine-learning methods.

In summary, we propose the descriptors based on the cohesive energy and atomic radius to characterize the bond strength of metals, and establish the quantitative structure-property relationship from bond strength, to GB energies, and to macro strength. Our descriptors, deriving from the d-band characteristics and broken-bond mechanism of tight-binding models, are applicable to not only the pure metals (including TMs and MGMs), but also the HEAs and BMGs, showing good prediction accuracy and wide applicability with reference to the experimental and theoretical results. This framework reveals physical pictures that the repulsive (attractive) effects play the kernel role in the variation of the bond-strength related properties for TMs and HEAs (MGMs and BMGs), which thus elucidates the role of elemental compositions, lattice structures, high-entropy effects, and amorphous effects in determining the strength of metals. Our scheme also reflects the heat of fusion, interstitial electron density, and inscribed sphere radius of the electron gas, exhibiting a robust physical basis. These findings not only serve as the solid physical guidance for understanding the relationship between bond strength and macro properties from the atomic-level perspective, but also provide an effective tool (also applicable to combine with machine-learning approaches) for accelerating the design of alloys with high performance, especially for MPEAs.

## Methods

The GB energy ($E_{GB}$) is defined as:

$$E_{GB} = \frac{E_{GB,\,system} - n_{GB}E_{bulk}}{2A_{GB}} \tag{10}$$

where $E_{GB,system}$ and $n_{GB}$ are the total energy and number of atoms of the GB structure, respectively, $A_{GB}$ is the cross-sectional area of the GB, $E_{bulk}$ is the energy per atom of the bulk. The strengthening energy of triple-junction GBs is defined as:

$$E_{str} = \frac{E_{GB,\,system} - E_{surf,\,1} - E_{surf,\,2} - E_{surf,\,3}}{A_{GB}} \tag{11}$$

where $E_{GB,system}$ is the total energy of the triple-junction GB structure. $E_{surf,1}$, $E_{surf,2}$ and $E_{surf,3}$ are the three separate surface structures that form the triple-junction GB structure. $A_{GB}$ is the cross-sectional area of the triple-junction GBs.

All the DFT calculations are performed by using the Vienna Ab initio Simulation Package (VASP)[53] with the projector augmented wave (PAW) method[54] and Perdew-Berke-Ernzerh (PBE) of functional[55]. The cutoff energy of 400 eV is used in the calculations and the k-point mesh is given in Supplementary Table 2. For the optimization, all the atoms are relaxed until the force on each of them is <0.02 eV/atom in our calculations. The calculations of Fe and Ni are spin-polarized. The twist and tilt GB-energy data can be found in ref. 19. The structures and details of asymmetric GBs can be found in Supplementary Fig. 4 and Supplementary Table 2.

All the macro-strength data used in our studies are measured by experiments. The bulk modulus $B$, ultimate strength $\tau_{max}$ and melting point temperature $T_m$ of pure metals are from refs. 9,14,37,56. The Young's moduli of HEAs are collected from ref. 42, and the Young's and

shear moduli, glass-transition temperature $T_g$ and the fracture strength of BMGs are collected from ref. 5.

## Data availability

All the data related to this work are collected by the literatures. All the data generated in this study are provided in the Source Data file. Data are also available from the corresponding author upon request. Source data are provided with this paper.

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

## Acknowledgements

The authors are thankful for the support from the National Natural Science Foundation of China (Nos. 22173034 and 11974128 for W. G., and 52130101 for Q. J.), the Opening Project of State Key Laboratory of High Performance Ceramics and Superfine Microstructure (SKL202206SIC for W. G.), the Program of Innovative Research Team (in Science and Technology) in University of Jilin Province, the Program for JLU (Jilin University) Science and Technology Innovative Research Team (No. 2017TD-09 for Q. J.), the Fundamental Research Funds for the Central Universities for W. G. and Q. J., and the computing resources of the High Performance Computing Center of Jilin University, China.

## Author contributions

W.G. and Q.J. conceived the original idea and designed the strategy. X.L. collected the data and performed the calculations. W.G. derived the models and analyzed the results with the contribution from X.L. X.L. and W.G. wrote the manuscript. X.L. prepared the Supplementary Information and drew all figures with the contribution from H.W. All authors have discussed and approved the results and conclusions of this article.

## Competing interests

The authors declare no competing interests.
