## [Transparent Peer Review file · Nature Communications]

A Roadmap from the Bond Strength to the Grain-Boundary Energies and Macro Strength of Metals

Corresponding Author: Professor Wang Gao

Version 0:

Reviewer comments:

Reviewer #1

(Remarks to the Author)

The paper aims to correlate bond strength with the strength of metals through a predictive model that determines grain-boundary energies and metal strength from cohesive energy and atomic radius. While this model is intriguing and the findings could potentially contribute to the field of multi-principal-element alloys (MPEAs), several critical issues must be addressed before considering publication.

1. Coordination Numbers: The definitions of coordination numbers of atoms on surfaces and grain boundaries are inadequately addressed. This is a significant oversight, particularly for complex surface and grain boundary structures, where precise coordination is essential for accurate modeling.
2. Grain Boundary Types: The study focuses on simple grain boundaries based on the coincident site lattice model. However, there are numerous types of grain boundaries due to various macroscopic and microscopic degrees of freedom. A more comprehensive approach considering this diversity is necessary for a robust model.
3. Bulk-Metallic Glasses (BMGs): The macro-mechanical properties of BMGs are highly sensitive to cooling rates, a factor not accounted for in the proposed descriptors. Without incorporating kinetic information, the model's predictions for BMGs are questionable.
4. Writing and Definitions: The manuscript needs substantial improvements in clarity and precision. Abbreviations should be defined when first introduced, and all terms, such as "Wd" in equation 3, must be explicitly explained.
5. Cohesive Energies and Atomic Radius: The paper does not provide sufficient details about the definitions and numerical values of cohesive energies and atomic radius used in the model. This lack of transparency hampers reproducibility and validation of the results by other researchers.
6. Scientific Discovery: The conclusion that strength-related material properties depend on cohesive energies and atomic sizes is not particularly surprising. These parameters are well-known to influence bond strengths and strain effects, thus diminishing the novelty and critical impact of the findings.

Given these significant issues, I recommend the rejection of the manuscript in its current form. The authors should address these concerns comprehensively to enhance the scientific rigor and clarity of their work.

Reviewer #2

(Remarks to the Author)

The authors present data showing a correlation between cohesive (bond) strength, grain boundary energy, bulk modulus, interstitial charge density, and shear strength that is shown to be generally applicable to many pure metals (FCC, BCC, and HCP) and alloys, including metallic glasses and concentrated random solid solutions. The concept is compelling, though key questions (see below) require attention before the paper can be accepted for publication.

A central problem is that it remains unclear to the reader how an alloy designer should select the appropriate power law dependence of these parameters with atomic radius (r_0), and what new insight is gained from the different atomic size dependencies. There appears to be a need to know a-priori whether 2.5 or 5 is more appropriate (eqs 4 and 5) for a given alloy, which is problematic for two reasons: (1) as an alloy designer it is not possible to reliably decide which correlation to use for composition selection, and (2) the choice between exponents seems arbitrary and disconnected from physical

meaning, i.e., a fitted rather than predictive one.

For example, the statement is made, "We find that the GB energies can be better described by E_{coh}/r_0^5 for BCC TMs but by $E_{coh}/r_0^{2.5}$ for the BCC MGMs (Figures 1a and b and Supplementary Figure 1), regardless of tilt or twist GBs and the GB orientations¹⁹, which are in good agreement with the prediction of our model (Eqs. 4 and 5)." Later, the authors comment that "Notably, the descriptor E_{coh}/r_0^5 scales linearly with ρ_0 for the TMs, whereas $E_{coh}/r_0^{2.5}$ describes the ρ_0 of the MGMs more accurately (Figure 2a)." So, is there a reason not to use the generally-valid interstitial charge density (ρ_0) instead of relying on what appears to be a fitting parameter (i.e., the power law factor based on atomic radius) that appears to vary depending on composition? In other words, the authors need to explain why ρ_0 , which is physically meaningful and easy to find (tabulated and calculated via rule-of-mixtures for alloys, per the cited works), is not as useful as the proposed $r^{2.5}$ or r^5 dependencies.

For example, do E_{coh} , E_{GB} , or stiffness (B or E) correlate better to ρ_0 for all cases if one scales ρ_0 by atomic volume, e.g., ρ_0/Ω ?

It is also well known that structural disorder changes the modulus of pure and alloyed metals (e.g., see Liu et al., "Nanocrystalline gold with small size: inverse Hall–Petch between mixed regime and super-soft regime", *Phil. Mag*, 2020). This can be associated more so with decreasing bond density or count than average bond strength (although these are interdependent). Can the authors explain how the structural disorder dependence of stiffness moduli in a pure metal like Au is connected to the proposed atomic-size-dependent correlation to modulus?

The authors use MPEA throughout the manuscript but then switch to HEAs in the caption of Figure 3. Recommend using MPEA or HEA everywhere, especially if a distinction is not explained. The use of "main group metals" and "main group elements" may be erroneous or at least confusing due to differences in how this term has been historically applied. It is recommended that the authors include a comment where they more clearly define the acronym "MGM" to say, "(MGMs, i.e., s- and p-block elements)".

It is unclear if the authors are accounting for the difference in atomic volume that occurs when multiple atoms are alloyed, which is relevant to both metallic glasses and concentrated solid solutions. For example, Al and Fe are FCC and BCC in pure form at room temperature, but their alloys can form crystalline structures (in HEAs) or clusters (in glasses) that impart very different atomic size. Which atomic sizes (or volumes) should one use when dealing with alloys and lattice structures different than those in pure, ambient temperature form? Here again the reviewer wonders if interstitial charge density is a more accurate correlation parameter than atomic radius.

From the right side of page 4 to the left side of page 5, the authors are explain why descriptors $E_{coh}/r_0^{2.5}$ and E_{coh}/r_0^5 are more applicable for MGMs and TMs, respectively. The statements are difficult to follow, especially for experimentalists. For example, they state, "For TMs, if only the repulsive part of the bond energies is considered in Eq. (7), one can derive $B \propto r_0^{-9}$ based on both Eqs (6) and (7), in consistent with the derivation from tight-binding theories and our descriptors E_{coh}/r_0^5 ." Why is it reasonable to only consider the repulsive part of the bond energies to arrive at a justification for E_{coh}/r_0^5 ? Why can the attractive component be neglected, especially considering the parameters n , m , c_1 , and c_2 in Eq 7 are fitting parameters in a Lennard-Jones potential which is a simplification with limited bounds of validity? Clearer explanation is needed on these key points.

Reviewer #3

(Remarks to the Author)

Reviewer #4

(Remarks to the Author)

The authors have presented a predictive model to estimate GB energy and strength of metals using cohesive energy and atomic radii. Transition metals and main group metals have been studied with a variety of GB types like $\Sigma 5$, $\Sigma 7$, and $\Sigma 9$. The expressions proposed for the quantities fit well with the GB energy data in literature for most metals. In my perspective the work is premature and falls below the high standards of nature comm. It does not have much novelty to offer and there are other details that will be presented in the points below.

The GB energy prediction has been well investigated in literature by using Machine learning and modeling. In the age of fast computation using high performance computing, there is no audience for empirical prediction using guessed modification of established expressions. This work fits the genre of 1960s and is unfortunately not applicable to the community today. The fits are not really that great. Some R2 values are in the range of 60s and 70s (supplementary). How can one really trust this predictive model. It is not a reliable model for material development. In the era of machine learning and computing, the authors are proposing an empirical formulation in a high impact journal. It will be unacceptable to consider this premature work. A simple search shows some excellent machine learning and modeling works done in the past: Please read: Restrepo, Sebastián Echeverri, Simón Tamayo Giraldo, and Barend J. Thijsse. "Using artificial neural networks to predict

grain boundary energies." Computational materials science 86 (2014): 170-173.

Waters, Brendon, Daniel S. Karls, Ilya Nikiforov, Ryan S. Elliott, Ellad B. Tadmor, and Brandon Runnels. "Automated determination of grain boundary energy and potential-dependence using the OpenKIM framework." Computational Materials Science 220 (2023): 112057.

Tamura, Tomoyuki, Masayuki Karasuyama, Ryo Kobayashi, Ryuichi Arakawa, Yoshinori Shiihara, and Ichiro Takeuchi. "Fast and scalable prediction of local energy at grain boundaries: machine-learning based modeling of first-principles calculations." Modelling and Simulation in Materials Science and Engineering 25, no. 7 (2017): 075003.
The present work is premature in the sense that it can be couple with more rigorous machine learning modeling to obtain a more reliable model with a mathematical function that can predict GB energy with higher accuracy.
The present work cannot be accepted.

Version 1:

Reviewer comments:

Reviewer #1

(Remarks to the Author)

The authors have made revisions in response to previous comments, which have improved the manuscript. However, I still feel that it does not yet reach the standard required for publication in Nature Communications. A primary concern is that grain boundaries (GBs) are inherently complex defects with atomistic structures and chemical compositions that depend on both thermodynamic and kinetic factors. While the authors identified empirical correlations between certain electronic/atomistic descriptors and simple grain boundary or metallic glass structures generated in simulations, real GB structures or metallic glass structures are far more intricate. A recent example illustrating this complexity is provided in the Science article, "Topological grain boundary segregation transitions" (Science 386 (2024) 420-424). Given these considerations, I believe the impact of this study remains limited, and I recommend that the manuscript not be accepted for publication at this time.

Reviewer #2

(Remarks to the Author)

The authors have addressed concerns from this reviewer, the manuscript is recommended for publication.

Reviewer #3

(Remarks to the Author)

Comments:

The paper aims to correlate bond strength with the strength of metals through a predictive model that determines grain-boundary energies and metal strength from cohesive energy and atomic radius. While this model is intriguing and the findings could potentially contribute to the field of multi-principal-element alloys (MPEAs), several critical issues must be addressed before considering publication.

Answer:

We thank the reviewer for showing great interest in our work and for the affirmations on the value and significance of our work. We also thank the reviewer for the efforts on our manuscript and for offering us constructive criticism that helped us improve our manuscript. According to the comments, we have carefully revised our manuscript, by presenting the details of our descriptors, the general application of these descriptors to complex grain boundaries (GBs), and highlighting the progress of our model. We hope that the revised manuscript will be judged appropriate for publication in *Nature Communications*.

Query 1:

1. Coordination Numbers: The definitions of coordination numbers of atoms on surfaces and grain boundaries are inadequately addressed. This is a significant oversight, particularly for complex surface and grain boundary structures, where precise coordination is essential for accurate modeling.

Answer 1:

We thank the reviewer for bringing up this issue. The coordination number of a given atom at surfaces and GBs is the sum of the weights of its nearest neighbors (that is obtained by partitioning their bond length with the bond length of bulk). The nearest neighboring atoms are classically identified by the collision detector of spheres, which calculates the pairs of spheres that overlap for a given atom. The test allows us to adopt the cutoff radius being 1.1 times of the shortest atomic distance in a pristine bulk structure, considering the deformation of the surface and GB region compared with the bulk. These approaches are widely used in determining the coordination number. Please see the main text on Page 2.

Query 2:

2. Grain Boundary Types: The study focuses on simple grain boundaries based on the coincident site lattice model. However, there are numerous types of grain boundaries due to various macroscopic and microscopic degrees of freedom. A more comprehensive approach considering this diversity is necessary for a robust model.

Answer 1:

We thank the reviewer for this insightful question. We calculated the energies of more complex GBs, such as the asymmetric (221)/(110), (411)/(100), (710)/(110), and (1130)/(970) GBs of FCC and BCC metals and the triple-junction GBs. Our descriptors are also capable of determining the stability of these GBs. All these results show that our descriptors have broad applicability to various types of GBs. Please see Supplementary Figure S3 and the main text on Page 3.

Query 3:

3. Bulk-Metallic Glasses (BMGs): The macro-mechanical properties of BMGs are highly sensitive to cooling rates, a factor not accounted for in the proposed descriptors. Without incorporating kinetic information, the model's predictions for BMGs are questionable.

Answer 3:

We thank the reviewer for the constructive comments. We acknowledge that the cooling rates can affect the mechanical properties of metallic glasses. Generally, only very high cooling rates (that can be over 10^{12} k/s) can lead to a sufficient structural change that generates a significant change in the plastic deformation of metallic glasses as proven in Ref. [5, 50, 51]. Accordingly, the resulting metallic glasses generally exhibit a small size (< 1 mm). In contrast, BMGs are usually obtained under relatively lower cooling rates. The BMGs of the used data reach the millimeter level of > 3 mm, with the cooling rates on the order of $< 10^3$ k/s, which have a minor influence on glass-transition temperature, Young's modulus, and fracture strength (see Ref. [5, 50, 51]). Notably, the glass-transition temperature is usually taken as a material characteristic, and correspondingly, Young's modulus and fracture strength can also be considered as the material characteristic under the given conditions (see Ref. [5]).

This study focuses on establishing the quantitative correlation between bond strength and the strength of BMGs (at the conditions being the material characteristic). The kinetic properties such as cooling-rate effects are another topic in the field of metallic glasses, which is clearly out of the scope of this study and could be our further topic. We have explained it on Page 7.

Query 4:

4. Writing and Definitions: The manuscript needs substantial improvements in clarity and precision. Abbreviations should be defined when first introduced, and all terms, such as "Wd" in equation 3, must be explicitly explained.

Answer 4:

We thank the reviewer for the suggestion. We have revised our manuscript carefully, by polishing the language and explaining the abbreviations and other parameters (including the W_d in the main text on Page 2).

Query 5:

5. Cohesive Energies and Atomic Radius: The paper does not provide sufficient details about the definitions and numerical values of cohesive energies and atomic radius used in the model. This lack of transparency hampers reproducibility and validation of the results by other researchers.

Answer 5:

We thank the reviewer for constructive criticism. We have elaborated on the details of cohesive energy and atomic radius as well as the references in the main text on Pages 2 and 3, and provide the numerical values in Supplementary Table 1.

Query 6:

6. *Scientific Discovery: The conclusion that strength-related material properties depend on cohesive energies and atomic sizes is not particularly surprising. These parameters are well-known to influence bond strengths and strain effects, thus diminishing the novelty and critical impact of the findings.*

Answer 6:

We thank the reviewer for the constructive comments. It has been a long-standing challenge to directly correlate the micro bond strength and macro strength of metals, prohibiting the fast prediction of macro strength properties. The cohesive energy and atomic radius are recognized to influence the bond strength; however, neither cohesive energy nor atomic radius can determine the strength-related properties alone. Quantitative prediction can only be achieved through the combination of these two parameters into a descriptor E_{coh}/r_0^5 for TMs and $E_{\text{coh}}/r_0^{2.5}$ for MGMs, as we found. We derive these bond-strength descriptors to quantify the macro strength from the d-band characteristics and broken-bond mechanism of tight-binding models, which are compatible with the classical model of bulk modulus, demonstrating the solid physical basis of our descriptors. This **quantitative and predictive framework of the strength-related properties of metals addresses the long-standing unresolved issue in the fields.**

Moreover, **our scheme uncovers novel physical pictures:** the repulsive effects play the dominant roles in the macro strength of metals with strong bond strength, such as TMs and HEAs, while the attractive effects dominate the macro strength of metals with the weak bond strength and the large atomic relaxation, like MGMs and BMGs, thus elucidating the role of elemental compositions, lattice structures, high-entropy effects, and amorphous effects in determining the strength of metals. Our physical-based descriptors can serve as useful tools for the design of advanced alloys. Our findings and their potential importance have been endorsed by the reviewers #2 and #3. We have revised the Introduction and Discussion section to highlight the novelty of our findings.

Comments:

The authors present data showing a correlation between cohesive (bond) strength, grain boundary energy, bulk modulus, interstitial charge density, and shear strength that is shown to be generally applicable to many pure metals (FCC, BCC, and HCP) and alloys, including metallic glasses and concentrated random solid solutions. The concept is compelling, though key questions (see below) require attention before the paper can be accepted for publication.

Answer:

We thank the reviewer for showing great interest in our work and for the full affirmations on the value and significance of our work. We also thank the reviewer for the efforts on our manuscript and for offering us constructive criticism that helped us improve our manuscript. According to the comments, we have revised the manuscript carefully, by comprising our descriptors with the previous ones and highlighting the physical meaning of our descriptors. We hope that the revised manuscript will be judged appropriate for publication in *Nature Communications*.

Query 1:

A central problem is that it remains unclear to the reader how an alloy designer should select the appropriate power law dependence of these parameters with atomic radius (r_0), and what new insight is gained from the different atomic size dependencies. There appears to be a need to know a-priori whether 2.5 or 5 is more appropriate (eqs 4 and 5) for a given alloy, which is problematic for two reasons: (1) as an alloy designer it is not possible to reliably decide which correlation to use for composition selection, and (2) the choice between exponents seems arbitrary and disconnected from physical meaning, i.e., a fitted rather than predictive one.

For example, the statement is made, “We find that the GB energies can be better described by E_{coh}/r_0^5 for BCC TMs but by $E_{coh}/r_0^{2.5}$ for the BCC MGMs (Figures 1a and b and Supplementary Figure 1), regardless of tilt or twist GBs and the GB orientations¹⁹, which are in good agreement with the prediction of our model (Eqs. 4 and 5).” Later, the authors comment that “Notably, the descriptor E_{coh}/r_0^5 scales linearly with ρ_0 for the TMs, whereas $E_{coh}/r_0^{2.5}$ describes the ρ_0 of the MGMs more accurately (Figure 2a).” So, is there a reason not to use the generally-valid interstitial charge density (ρ_0) instead of relying on what appears to be a fitting parameter (i.e., the power law factor based on atomic radius) that appears to vary depending on composition? In other words, the authors need to explain why ρ_0 , which is physically meaningful and easy to find (tabulated and calculated via rule-of-mixtures for alloys, per the cited works), is not as useful as the proposed $r^{2.5}$ or r^5 dependencies. For example, do E_{coh} , E_{GB} , or stiffness (B or E) correlate better to ρ_0 for all cases if one scales ρ_0 by atomic volume, e.g., ρ_0/Ω ?

Answer 1:

We thank the reviewer for the constructive comments.

1) The determination and physical meaning of the power law

First, **the power law of the atomic radius in our descriptors is certain:** 5 for transition metals and high entropy alloys (HEAs), and 2.5 for main-group metals (MGMs) and bulk metallic glasses (BMGs). Moreover, the power law of the atomic radius in our descriptors is derived from the d-band characteristics and broken-bond mechanism of tight-binding models, which is not a fitting parameter. Furthermore, **the different power laws of the atomic radius uncover a novel physical picture:** the

repulsive effects dominate the macro strength of the metals with strong bonds, like TMs and HEAs, while the attractive effects do the macro strength of the metals with weak bonds, like MGMs and BMGs, thus elucidating the role of elemental compositions, lattice structures, high-entropy effects, and amorphous effects in determining the strength of metals. Please see the section of the Results on Page 2 and the revision on Page 8.

2) The meaning of our descriptors

The construction of the structure-property relationship between bond strength and macro strength has been a long-standing challenge in the fields. **Our bond-strength descriptors resolve this issue successfully**, which has never been achieved by other descriptors, such as the interstitial electron density that is only indirectly related to the bond strength. Our descriptors are derived from the d-band characteristics and broken-bond mechanism of tight-binding models, and are compatible with the classical model of bulk modulus, demonstrating the solid physical basis of our descriptors. We have revised Introduction and Discussion to further highlight the meaning and novelty of our descriptors.

3) The comparison of our descriptors with the previous studies.

From the perspective of accuracy, our descriptors are more accurate than the previous ones. For the prediction of GB energies, the regression coefficient is 0.84 for our descriptors compared with 0.76 for ρ_0 (Supplementary Figure 7b). For the prediction of the elastic modulus E of HEAs, the regression coefficient is 0.86 for our descriptors (Figures. 3a and 3c), but about 0.67 for ρ_0 (see Figure 5 of Ref.[13]) and 0.74 for ρ_0/Ω (Supplementary Figure 7c). Moreover, ρ_0 requires time-burdensome numerical calculations, the data of which are still unavailable for some elements such as C, Si, and rare-earth elements. In comparison, the values of cohesive energy and atomic radius in our descriptors cover almost all the elements, and are all accessible by table looking up, which reduces the uncertainty of numerical calculations. We have presented the comparison of our descriptors with the previous ones in the main text on Page 8.

Overall, our physical-based descriptors provide not only a predictive tool for material design but also new insights into the structure-property relationship between bond strength and macro-strength properties of metals.

Query 2:

It is also well known that structural disorder changes the modulus of pure and alloyed metals (e.g., see Liu et al., “Nanocrystalline gold with small size: inverse Hall–Petch between mixed regime and super-soft regime”, Phil. Mag, 2020). This can be associated more so with decreasing bond density or count than average bond strength (although these are interdependent). Can the authors explain how the structural disorder dependence of stiffness moduli in a pure metal like Au is connected to the proposed atomic-size-dependent correlation to modulus?

Answer 2:

We thank the reviewer for the constructive comments. With the decrease of the grain size of a pure metal like Au, the number of atoms at GBs increases and thus the metal structures tend to be disorder. Accordingly, the bond density of the metal reduces, leading to the decrease of the interatomic bond strength (E_{coh}). Therefore, the value of the descriptor (E_{coh}/r_0^5) turns to decrease and thus the modulus

of the metal gets smaller according to our framework, which is consistent with the results in the suggested reference (Ref.[52]). Clearly, our framework can provide a qualitative interpretation of the structural disorder dependence of moduli in a pure metal like Au. We have explained it in the main text on Page 7.

Query 3:

The authors use MPEA throughout the manuscript but then switch to HEAs in the caption of Figure 3. Recommend using MPEA or HEA everywhere, especially if a distinction is not explained.

Answer 3:

We thank the reviewer for this valuable suggestion. The adopted HEAs are a type of MPEAs, and the used BMGs are in the form of both binary alloys and MPEAs. We have carefully revised these descriptions in the main text on Page 6 and elsewhere.

Query 4:

The use of “main group metals” and “main group elements” may be erroneous or at least confusing due to differences in how this term has been historically applied. It is recommended that the authors include a comment where they more clearly define the acronym “MGM” to say, “(MGMs, i.e., s- and p-block elements)”.

Answer 4:

We thank the reviewer for this valuable suggestion. The main-group metals (MGMs) include Li, Na, K, Rb, Cs, Be, Mg, Ca, Sr, Ba, and Al (totally 11 metals) in this work, and the main-group elements contain MGMs and other s- and p-block elements like Si, C, and Sn (which are the components in studied HEAs and BMGs). We have carefully elaborated on these descriptions in the main text on Pages 3 and 6.

Query 5:

It is unclear if the authors are accounting for the difference in atomic volume that occurs when multiple atoms are alloyed, which is relevant to both metallic glasses and concentrated solid solutions. For example, Al and Fe are FCC and BCC in pure form at room temperature, but their alloys can form crystalline structures (in HEAs) or clusters (in glasses) that impart very different atomic size. Which atomic sizes (or volumes) should one use when dealing with alloys and lattice structures different than those in pure, ambient temperature form? Here again the reviewer wonders if interstitial charge density is a more accurate correlation parameter than atomic radius.

Answer 5:

We thank the reviewer for constructive comments. We agree that there are differences in atomic volume during alloying. However, HEAs may have mixture phases (such as BCC+FCC, FCC+intermetallics), and BMGs exhibit complicated geometric structures due to the structural disorder, making the radius choice based on the alloy phases not unique. Nevertheless, the differences of atomic sizes in different phases cause negligible effects on the predictive power of our descriptors (Figure 3a and Supplementary Figure 7a). Therefore, **we simply use the atomic radii obtained from the most stable crystal or molecule structure**, defined as half of the length of a bond in the most stable crystal structure or molecule, which are easily obtained by table looking up. For example, only

the radius of an atom in the BCC structure is used for Fe. Moreover, our descriptors predict the macro strength of MPEAs more accurately than the interstitial charge density (see the answer of query 1). We have added these descriptions in the main text on Page 6.

Query 6:

From the right side of page 4 to the left side of page 5, the authors are explain why descriptors $E_{coh}/r^{0.5}$ and E_{coh}/r^5 are more applicable for MGMs and TMs, respectively. The statements are difficult to follow, especially for experimentalists. For example, they state, “For TMs, if only the repulsive part of the bond energies is considered in Eq. (7), one can derive $B \propto r^{-9}$ based on both Eqs (6) and (7), in consistent with the derivation from tight-binding theories and our descriptors E_{coh}/r^5 .” Why is it reasonable to only consider the repulsive part of the bond energies to arrive at a justification for E_{coh}/r^5 ? Why can the attractive component be neglected, especially considering the parameters n , m , $c1$, and $c2$ in Eq 7 are fitting parameters in a Lennard-Jones potential which is a simplification with limited bounds of validity? Clearer explanation is needed on these key points.

Answer 6:

We thank the reviewer for bringing up this issue. From the perspective of bond energy, the attractive part ($E_{coh, attr}$) always dominates the cohesive energy at equilibrium and thus one can derive $B \propto E_{coh, attr}/r^{2.5} \propto r^{-3.5}$ for MGMs and $B \propto E_{coh, attr}/r^5 \propto r^{-9}$ for TMs according to the tight-binding theories and our descriptors. From the perspective of atomic interaction force, the repulsive and attractive forces are equal at equilibrium. The classical model of bulk modulus for MGMs only considers the contribution of the attractive force, and thus derives the relationship $B \propto -\Omega_0 \left(\frac{\partial \sigma}{\partial \Omega} \right)_{\Omega_0} \propto (\Omega_0)^{-4/3} \propto r_0^{-4}$. In contrast, the repulsive force plays a more important role in TMs than in MGMs (as the bond length of TMs is obviously smaller than that of MGMs) and thus one can derive $B \propto -\Omega_0 \left(\frac{\partial \sigma}{\partial \Omega} \right)_{\Omega_0} \propto r_0^{-9}$ for TMs from Eqs. 6 and 7 based on the expression of the repulsive force. The resulting decay behavior from both the energy and force derivations are consistent, regardless of MGMs or TMs, demonstrating the conclusion that the attractive and repulsive forces dominate the trends of the modulus of MGMs and TMs, respectively. We have revised the misleading statements and explained how our descriptors are compatible with the classical theories on Page 5 of the main text in order to better understand our framework.

Comments:

The authors have presented a predictive model to estimate GB energy and strength of metals using cohesive energy and atomic radii. Transition metals and main group metals have been studied with a variety of GB types like $\Sigma 5$, $\Sigma 7$, and $\Sigma 9$. The expressions proposed for the quantities fit well with the GB energy data in literature for most metals.

In my perspective the work is premature and falls below the high standards of nature comm. It does not have much novelty to offer and there are other details that will be presented in the points below.

Answer:

We thank the reviewer for the comments and have carefully addressed all comments to improve our manuscript. It has been a long-standing challenge to develop physics-based metrics to correlate the bond strength and macro strength of metals, which is crucial to designing high-performance materials and understanding the underlying mechanism. We propose the bond-strength descriptors, derived from the d-band characteristics and broken-bond mechanism of tight-binding models, which can accurately determine the strength-related properties. Compared with machine-learning approaches, our framework not only exhibits the solid physical origin, but also unravels a novel physical picture between bond strength and macro strength. We hope that the revised manuscript will be judged appropriate for publication in *Nature Communications*.

Query 1&3:

The GB energy prediction has been well investigated in literature by using Machine learning and modeling. In the age of fast computation using high performance computing, there is no audience for empirical prediction using guessed modification of established expressions. This work fits the genre of 1960s and is unfortunately not applicable to the community today.

In the era of machine learning and computing, the authors are proposing an empirical formulation in a high impact journal. It will be unacceptable to consider this premature work.

A simple search shows some excellent machine learning and modeling works done in the past: Please read:

*Restrepo, Sebastián Echeverri, Simón Tamayo Giraldo, and Barend J. Thijsse. "Using artificial neural networks to predict grain boundary energies." *Computational materials science* 86 (2014): 170-173.*

*Waters, Brendon, Daniel S. Karls, Ilia Nikiforov, Ryan S. Elliott, Ellad B. Tadmor, and Brandon Runnels. "Automated determination of grain boundary energy and potential-dependence using the OpenKIM framework." *Computational Materials Science* 220 (2023): 112057.*

*Tamura, Tomoyuki, Masayuki Karasuyama, Ryo Kobayashi, Ryuichi Arakawa, Yoshinori Shiihara, and Ichiro Takeuchi. "Fast and scalable prediction of local energy at grain boundaries: machine-learning based modeling of first-principles calculations." *Modelling and Simulation in Materials Science and Engineering* 25, no. 7 (2017): 075003.*

Answer 1:

We thank the reviewer for the comments.

i) Developing the physics-based metrics for designing high-performance materials has always been a core goal in material science, regardless of the era. Indeed, the study of the development of analytic descriptors and models for predicting material properties is a subject of considerable interest to the

readership of Nature Series, as judged for example by the following literature (most likely an incomplete reference list).

- 1) Thermodynamic stability of ligand-protected metal nanoclusters. *Nat. Commun.* **8**, 15988 (2017);
- 2) High-entropy high-hardness metal carbides discovered by entropy descriptors. *Nat. Commun.* **9**, 4980 (2018);
- 3) Engineering atomic-level complexity in high-entropy and complex concentrated alloys. *Nat. Commun.* **10**, 2090 (2019);
- 5) Local electronic descriptors for solute-defect interactions in bcc refractory metals. *Nat. Commun.* **10**, 4484 (2019);
- 4) Screening highly active perovskites for hydrogen-evolving reaction via unifying ionic electronegativity descriptor. *Nat. Commun.* **10**, 3755 (2019);
- 6) Determining the adsorption energies of small molecules with the intrinsic properties of adsorbates and substrates. *Nat. Commun.* **11**, 1196 (2020);
- 7) Electronic parameters in cobalt-based perovskite-type oxides as descriptors for chemo-catalytic reactions. *Nat. Commun.* **11**, 652 (2020);
- 8) Predictive model of hydrogen trapping and bubbling in nanovoids in bcc metals. *Nat. Mater.* **18**, 833-839 (2019);
- 9) The importance of a charge transfer descriptor for screening potential CO₂ reduction electrocatalysts. *Nat. Commun.* **14**, 2598 (2023).
- 10) Theory-guided design of high-entropy alloys with enhanced strength-ductility synergy. *Nat. Commun.* **14**, 2519 (2023);

In particular, it has been a long-standing unresolved issue to directly correlate the micro bond strength and macro strength of metals, prohibiting the fast prediction of macro strength properties. Machine-learning-based approaches are the potential ways to predict the GB energies and macro strength, however, it is well known that they exhibit unclear interpretability and are black-box methods. For instance, neural networks, as used in the suggested literature 1, are generally with less physical insight and unclear mathematical functions. Some other approaches, such as LASSO models with different optimization and descriptors (as used in the suggested literature 3), still lack the concise mathematical function of GB energies (not to mention macro strength) with less physical insight. In addition, the energies predicted by the different interatomic potential calculations were found to correlate closely with the energy from a semi-analytic geometric model in the suggested literature 2, thereby the qualitative form of the GB energy versus tilt angle was dominated more by geometry than the choice of interatomic potential, indicating that the physics-based model is essential for predicting the GB energies. Therefore, identifying the physics-based determinants and structure-property relationships of strength-related properties is urgently needed in material science.

We derive the bond-strength descriptors to quantify the macro strength from the d-band characteristics and broken-bond mechanism of tight-binding models, which are compatible with the classical model of bulk modulus, demonstrating that **our descriptors exhibit explicit physical meanings**. Moreover, **our descriptors uncover the novel physical picture** of the macro strength of metals: the repulsive effects play the dominant role in the macro strength of metals with strong bond strength, such as TMs and HEAs, while the attractive effects dominate the macro strength of metals

with the weak bond strength and the large atomic relaxation, like MGMs and BMGs, thus elucidating the role of elemental compositions, lattice structures, high-entropy effects, and amorphous effects in determining the strength of metals. All these results overcome the drawbacks of the black-box machine-learning methods. In addition, our descriptors show good prediction accuracy and wide applicability with reference to both theoretical and experimental results. Our findings and their potential importance have been endorsed by the reviewers #1, #2 and #3.

ii) Moreover, **the machine-learning approaches strongly depend on the effective and accessible descriptors**, for which our descriptors could be choices. Therefore, we **provide the machine-learning models** by using the support-vector machine, random-forest regression, and gradient-boosting decision tree methods (in Supplementary Figure 8), which all predict the macro strength of HEAs and BMGs accurately but cannot be extracted any concise mathematical function. Notably, these machine-learning models are constructed with only one single feature (E_{coh}/r_0^5 for HEAs and $E_{\text{coh}}/r_0^{2.5}$ for BMGs), exhibiting a pronounced simplicity and demonstrating the advantage of physical-based descriptors. We have added the machine-learning results in the main text on Page 8 and Supplementary Figure 8.

Query 2:

The fits are not really that great. Some R2 values are in the range of 60s and 70s (supplementary). How can one really trust this predictive model. It is not a reliable model for material development.

Answer 2:

We thank the reviewer for the constructive comments. Most of the R^2 values of the relationship between GB energies/macro strength and our descriptors are above 0.85. The R^2 values are ~ 0.7 for determining the GB energies of main-group metals, since the ranges of GB energies of main-group metals are as low as ~ 0.1 J/m². These ranges are within the error bar of calculations of DFT semi-local functionals, which leads to the prediction error of the GB energies of main-group metals. Nevertheless, our descriptors can still determine the trend of the GB energies of MGMs even with such a small energy range.

Query 4:

The present work is premature in the sense that it can be couple with more rigorous machine learning modeling to obtain a more reliable model with a mathematical function that can predict GB energy with higher accuracy.

Answer 4:

We thank the reviewer for the comments. It is well known that machine-learning methods are black-box tools with no concise mathematical functions and less physical insight, as the answer 1 explained. Indeed, interpretability is one of the fundamental challenges for machine learning, not to mention the extraction of concise mathematical functions with physical insights from machine-learning models. We identify the physics-based descriptors, the bond-strength descriptors, for determining the macro strength of metals, which provide not only novel physical pictures for strength-related properties but also a predictive tool for material design. Our models have contributed significantly to correlating the micro bond strength and macro strength of metals, a long-term challenge in material science. We hope that the revised manuscript will be judged appropriate for publication in *Nature Communications*.

Comments:

The authors have made revisions in response to previous comments, which have improved the manuscript. However, I still feel that it does not yet reach the standard required for publication in Nature Communications. A primary concern is that grain boundaries (GBs) are inherently complex defects with atomistic structures and chemical compositions that depend on both thermodynamic and kinetic factors. While the authors identified empirical correlations between certain electronic/atomistic descriptors and simple grain boundary or metallic glass structures generated in simulations, real GB structures or metallic glass structures are far more intricate. A recent example illustrating this complexity is provided in the Science article, "Topological grain boundary segregation transitions" (Science 386 (2024) 420-424). Given these considerations, I believe the impact of this study remains limited, and I recommend that the manuscript not be accepted for publication at this time.

Answer:

We thank the reviewer for the efforts on our manuscript and for offering us constructive criticism that helped us improve our manuscript. We focus on the structure-property relationship across bond strength, grain-boundary (GB) energies, and macro strength of metals. Our scheme is applicable to determine the GB energies from simple symmetric GBs to complex asymmetric GBs and triple-junctions, and to determine the experimental macro strength of more complex real systems like high-entropy alloys (HEAs) and bulk metallic glasses (BMGs). We acknowledge that GB structures and metallic glass structures are intricate, nevertheless, the studied properties (e.g. GB energies and moduli) originate from the bond strength and can be quantified by our framework regardless of the structure complexity. Our scheme further identifies the role of elemental compositions, lattice structures, high-entropy, and amorphous effects in determining the strength of metals. The GB segregation and phase transitions that are studied in the referred literature are another important topics, out of the scope of this work, for which our framework could serve as a useful basis. We have explained on Page 8.

Report of Reviewer #2 -- NCOMMS-24-36957A-Z

Comments:

The authors have addressed concerns from this reviewer, the manuscript is recommended for publication.

Answer:

We appreciate the reviewer for the affirmations on the value and significance of our work, and for the recommendation for publication in *Nature Communications*.

Report of Reviewer #3 -- NCOMMS-24-36957A-Z

Comments:

Answer:

We appreciate the reviewer for the affirmations on the value and significance of our work, and for the recommendation for publication in *Nature Communications*.